# Learning identifiable and interpretable latent models of high-dimensional neural activity using pi-VAE

**Ding Zhou**
Department of Statistics
Columbia University
dz2336@columbia.edu

**Xue-Xin Wei**
Department of Neuroscience
UT Austin
weixx@utexas.edu

## Abstract

The ability to record activities from hundreds of neurons simultaneously in the brain has placed an increasing demand for developing appropriate statistical techniques to analyze such data. Recently, deep generative models have been proposed to fit neural population responses. While these methods are flexible and expressive, the downside is that they can be difficult to interpret and identify. To address this problem, we propose a method that integrates key ingredients from latent models and traditional neural encoding models. Our method, pi-VAE, is inspired by recent progress on identifiable variational auto-encoder, which we adapt to make appropriate for neuroscience applications. Specifically, we propose to construct latent variable models of neural activity *while simultaneously* modeling the relation between the latent and task variables (non-neural variables, *e.g.* sensory, motor, and other externally observable states). The incorporation of task variables results in models that are not only more constrained, but also show qualitative improvements in interpretability and identifiability. We validate pi-VAE using synthetic data, and apply it to analyze neurophysiological datasets from rat hippocampus and macaque motor cortex. We demonstrate that pi-VAE not only fits the data better, but also provides unexpected novel insights into the structure of the neural codes.

## 1   Introduction

Popular analysis methods of neural responses in neurophysiology mainly come in two classes: one based on regression, the other on latent variable modeling. Generalized Linear Model (e.g., (67; 56)) and tuning curve analysis (24; 73; 50) are notable examples of the regression-based approach, and both have been widely used in neuroscience in the past few decades (63; 60; 73; 54; 11). These methods express the neural firing rate as a function of the stimulus variable, thus naturally define encoding models, which can be inverted to decode the stimulus variables (24; 59; 73; 6; 51). In contrast, the latent-based approach aims to account to variability of the neural responses using a relatively small number of latent variables which are typically not observed. Recently, various latent-based methods have been developed or applied to analyze neural data and in particular simultaneously recorded neural population data, including principal component analysis (66; 46; 4; 13; 45), factor analysis (10; 58; 18), linear/nonlinear dynamical systems (44; 7; 22; 17; 52), among others (*e.g.*, (5; 74; 71; 34)). Each class of models carries certain advantages and disadvantages. Regression-based methods tend to have higher interpretability, however, they often suffer the problem of under-fitting. Latent-based models are more flexible in accounting for the neural variability, however they may be difficult to interpret and sometimes not identifiable. Notably, some studies had incorporated latent fluctuations into the encoding models (72; 39; 25; 19; 42; 11) yielding promising results, although these models often assumed highly specialized latent structure, thus potentially limits the applicability in practice.

The issues of identifiability and interpretability are becoming increasingly important as the neuroscience community adapts more sophisticated methods from nonlinear deep generative models (17; 70). Deep generative models have the promise of extracting complex nonlinear structure which may be difficult to achieve by linear methods, as demonstrated by recent work based on the variational auto-encoder (VAE) (*e.g.*, (37; 57; 22; 52)). However, over the past few years it has become increasingly clear that the latents extracted from these models, and VAE in particular, are often highly entangled therefore difficult to interpret (29; 28; 1; 12; 43; 8; 35). Given these considerations, an important question is how to model neural population responses with nonlinear models that are powerful yet scientifically insightful via identifiability and interpretability.

We propose a model formulation which represents one step toward addressing this question. Specifically, we draw on recent progress on identifiable VAE (iVAE) (35; 64), and generalize and adapt it to make it directly applicable to a broad variety of datasets. Conceptually, our method combines the respective strengths of regression-based and latent-based approaches (Fig. 1): i) by using the VAE architecture, it is expressive and flexible; ii) by treating task variables as labels and explicitly modeling them with the latents, our method is better constrained and under certain conditions identifiable. We apply our method to synthetic data and eletrophysiological datasets from population recording of rat hippocampus in a navigation task (27; 26) and the motor area of macaque during a reaching task (21). We demonstrate that our method can recover interpretable latent structure that is informative about the structure of the neural code and dynamics.

## 2 Model

**Notations** We denote $\mathbf{x} \in \mathbb{R}^n$ as observations. For our purpose, $\mathbf{x}$ specifically represents the population response or spike counts within a small time window. We use $\mathbf{u} \in \mathbb{R}^d$ to represent *task variables* (or *labels*), that are measured along with the neural activities, e.g., the location of the animal when studying navigation tasks. $\mathbf{u}$ can be discrete or continuous. Additionally, we denote $\mathbf{z} \in \mathbb{R}^m (m \ll n)$ as the unobserved low-dimensional latent variables.

### 2.1 Generative model

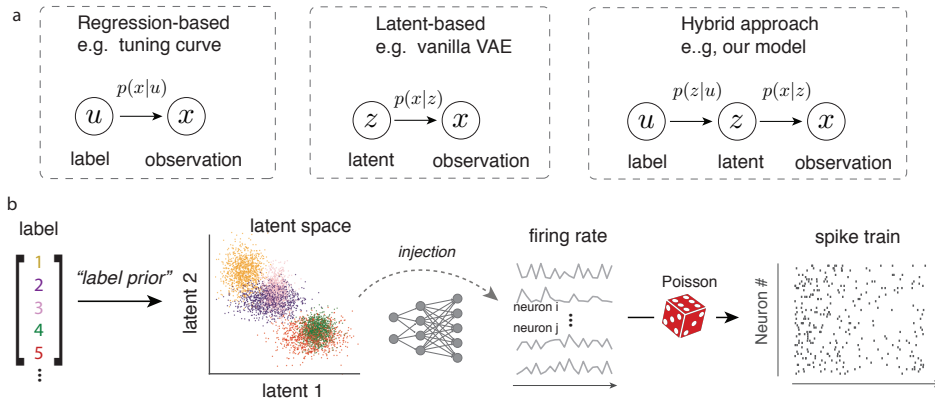

Figure 1: The model framework and generative model. (a) Structure of three classes of statistical models for neural data analysis. Our method is based on the integration of the first two classes into a hybrid approach. Our approach models the statistical dependence between label ($\mathbf{u}$) and latent ($\mathbf{z}$) as well as between latent ($\mathbf{z}$) and observation ($\mathbf{x}$) simultaneously. (b) Schematic illustration of the generative model of pi-VAE. Major components include the "label prior" between the task variables and the latent, an injective mapping between latent and firing rate parameterized by normalizing flow, and Poisson observation noise.

Our goal is to develop models that are flexible and expressive in capturing the variability of the data, while also well-constrained so that the models would enjoy identifiability and interpretability. Motivated by these considerations, we propose a generative model formulation which integrates key ingredients of the latent-based and regression-based approaches (see Fig. 1a):

$$p_{\boldsymbol{\theta}}(\mathbf{x}, \mathbf{z}|\mathbf{u}) = p_{\mathbf{f}}(\mathbf{x}|\mathbf{z})p_{\mathbf{T},\boldsymbol{\lambda}}(\mathbf{z}|\mathbf{u}). \tag{1}$$

This is a general formulation, and some previous models may be re-formulated to conform with it (*e.g.*, (39; 25; 19)). In this paper, we will focus on a specific implementation that is directly inspired by the recent work on identifiable VAE (35; 41; 64). In the interest of neuroscience applications, we have developed a method that can simultaneously deal with Poisson noise, both discrete and continuous labels, and larger output dimension than input dimension (Fig. 1b). We will show that our model is sufficiently expressive for many applications, yet still constrained enough to be identifiable.

We start by defining the component that describes the relation between the label and the latent, *i.e.*, $p_{\mathbf{T},\boldsymbol{\lambda}}(\mathbf{z}|\mathbf{u})$. We will refer to it as the *"label prior"*. Following (35), we assume $p_{\mathbf{T},\boldsymbol{\lambda}}(\mathbf{z}|\mathbf{u})$ to be conditionally independent, where each element $\mathbf{z}_i \in \mathbf{z}$ has *an exponential family distribution* given $\mathbf{u}$,

$$p_{\mathbf{T},\boldsymbol{\lambda}}(\mathbf{z}|\mathbf{u}) = \prod_{i=1}^{m} p(\mathbf{z}_i|\mathbf{u}) = \prod_{i=1}^{m} \frac{Q_i(\mathbf{z}_i)}{Z_i(\mathbf{u})} \exp\left[ \sum_{j=1}^{k} T_{i,j}(\mathbf{z}_i)\lambda_{i,j}(\mathbf{u}) \right], \tag{2}$$

where $Q_i$ is the base measure, $\mathbf{T}_i = (T_{i,1}, \cdots, T_{i,k})$ are the sufficient statistics, $Z_i(\mathbf{u})$ is the normalizing factor, $\boldsymbol{\lambda}_i = (\lambda_{i,1}(\mathbf{u}), \cdots, \lambda_{i,k}(\mathbf{u}))$ are the natural parameters, and $k$ is pre-defined number of sufficient statistics. Practically, a small number of components $k$ is often sufficient for the problems which we have considered. For discrete $\mathbf{u}$, we simply use a different $\lambda_{ij}$ for different $\mathbf{u}$. To deal with continuous $\mathbf{u}$, we develop a procedure by parameterizing $\lambda_{ij}$ as a function of $\mathbf{u}$ using a feed-forward neural network. Details are given in Supplementary Information (SI).

We next turn to the dependence between the latent and observation, *i.e.*, $p_{\mathbf{f}}(\mathbf{x}|\mathbf{z})$. In (35), $p_{\mathbf{f}}(\mathbf{x}|\mathbf{z})$ is defined using additive noise $p_{\mathbf{f}}(\mathbf{x}|\mathbf{z}) = p_{\boldsymbol{\epsilon}}(\mathbf{x} - \mathbf{f}(\mathbf{z}))$, i.e. $\mathbf{x} = \mathbf{f}(\mathbf{z}) + \boldsymbol{\epsilon}$, where $\boldsymbol{\epsilon}$ is an independent noise variable. To model the spike data, we generalize it to Poisson model $p_{\mathbf{f}}(\mathbf{x}|\mathbf{z}) = \text{Poisson}(\mathbf{f}(\mathbf{z}))$ with $\mathbf{f}$ being the instantaneous firing rate to deal with the count observations. We implement $\mathbf{f}$ using normalizing flow as detailed later. Putting together, we denote $\boldsymbol{\theta} = (\mathbf{f}, \mathbf{T}, \boldsymbol{\lambda})$ as parameters in generative model 1. We refer to our model as Poission identifiable VAE, or pi-VAE for simplicity.

**Identifiability** (35) has proved that the additive noise model is identifiable with certain assumptions. Under the same assumptions, we can prove that pi-VAE is also identifiable.

**Definition 1.** *Let $\sim$ be an equivalence relation on the domain of the parameters $\Theta = (\boldsymbol{\theta} := (\mathbf{f}, \mathbf{T}, \boldsymbol{\lambda}))$. Model 1 is said to be identifiable up to $\sim$ if $p_{\boldsymbol{\theta}}(\mathbf{x}|\mathbf{u}) = p_{\tilde{\boldsymbol{\theta}}}(\mathbf{x}|\mathbf{u}) \implies \boldsymbol{\theta} \sim \tilde{\boldsymbol{\theta}}$.*

**Definition 2.** *Define $\sim$ as $\boldsymbol{\theta} \sim \tilde{\boldsymbol{\theta}} \iff \exists A, c, \mathbf{T}\left(\mathbf{f}^{-1}(\mathbf{x})\right) = A\tilde{\mathbf{T}}\left(\tilde{\mathbf{f}}^{-1}(\mathbf{x})\right) + c, \forall \mathbf{x} \in \text{Img}(\mathbf{f}) \subset \mathbb{R}^n$, where $\boldsymbol{\theta} = (\mathbf{f}, \mathbf{T}, \boldsymbol{\lambda}), \tilde{\boldsymbol{\theta}} = (\tilde{\mathbf{f}}, \tilde{\mathbf{T}}, \tilde{\boldsymbol{\lambda}})$, $A$ is a full rank $mk \times mk$ matrix, $c \in \mathbb{R}^{mk}$ is a vector.*

**Theorem 1.** *Assume that we observe data sampled from pi-VAE model defined according to equation 1,2 with Poisson noise and parameters $\boldsymbol{\theta} = (\mathbf{f}, \mathbf{T}, \boldsymbol{\lambda})$. Assume the following holds:*

*i) The firing rate function $\mathbf{f}$ in equation 1 is injective,*

*ii) The sufficient statistics $T_{i,j}$ in 2 are differentiable almost everywhere, and their derivatives $T'_{i,j}$ are nonzero almost everywhere for $1 \leq i \leq m, 1 \leq j \leq k$,*

*iii) There exists $mk + 1$ distinct points $\mathbf{u}^0, \cdots, \mathbf{u}^{mk+1}$ such that the matrix*

$$L = \left( \boldsymbol{\lambda}(\mathbf{u}^1) - \boldsymbol{\lambda}(\mathbf{u}^0), \cdots, \boldsymbol{\lambda}(\mathbf{u}^{mk}) - \boldsymbol{\lambda}(\mathbf{u}^0) \right)$$

*of size $mk \times mk$ is invertible, then the pi-VAE model is identifiable up to $\sim$.*

This theorem is a straight-forward generalization of the results in (35). Proof is given in the SI. This theorem means, if two sets of model parameters lead to the same marginal distribution of $\mathbf{x}$, then $(\mathbf{f}, \mathbf{T}, \boldsymbol{\lambda}) \sim (\tilde{\mathbf{f}}, \tilde{\mathbf{T}}, \tilde{\boldsymbol{\lambda}})$. Thus one can hope to recover posterior distribution $p(\mathbf{z}|\mathbf{x})$ up to a linear transformation $A$ and point-wise non-linearities between $\mathbf{T}$ and $\tilde{\mathbf{T}}$, as well as the joint distribution $p(\mathbf{x}, \mathbf{z})$. Note that other forms of identifiability may be derived with modified assumptions (35). While practically, without knowing the ground truth, the assumptions may be difficult to verify, some encouraging preliminary evidence suggest that identifiability may have some robustness with mild violations of model assumptions (64).

We next describe how to parameterize the injection $\mathbf{f} : \mathbb{R}^m \to \mathbb{R}^n$. Here we extend the General Incompressible-flow Network (GIN) proposed in (64), which shares the flexibility of RealNVP (16) and the volume-preserving property of NICE (15). Practically, we found our implementation to be reasonably efficient computationally. Specifically, GIN defines a mapping from $\mathbb{R}^D \to \mathbb{R}^D$ with

Jacobian determinant equal to 1 (64). It splits the $D$-dimensional input $\mathbf{x}$ into two parts $\mathbf{x}_{1:l}, \mathbf{x}_{l+1:D}$, where $l < D$. The output $\mathbf{y}$ is defined as the concatenation of $\mathbf{y}_{1:l}$ and $\mathbf{y}_{l+1:D}$,

$$\mathbf{y}_{1:l} = \mathbf{x}_{1:l} \tag{3}$$
$$\mathbf{y}_{l+1:D} = \mathbf{x}_{l+1:D} \odot \exp\left(s\left(\mathbf{x}_{1:l}\right)\right) + t\left(\mathbf{x}_{1:l}\right), \tag{4}$$

where $s(\cdot)$ and $t(\cdot)$ are both functions defined on $\mathbb{R}^l \to \mathbb{R}^{D-l}$, and the total sum of $s\left(\mathbf{x}_{1:l}\right)$ is constrained to be zero by setting the final component to be the negative sum of previous components.

The original GIN (64) only deals with the case where the input and output have the same dimensions. In our case, the output dimension is often much larger than the input dimension. We thus develop a new scheme to parameterize $\mathbf{f} : \mathbb{R}^m \to \mathbb{R}^n$ which retains the properties of GIN. We first map $\mathbf{z}_{1:m}$ to the concatenation of $\mathbf{z}_{1:m}$ and $t\left(\mathbf{z}_{1:m}\right)$. Note that this is equivalent to GIN model with input as $\mathbf{z}_{1:m}$ padding $n$-$m$ zeros. We then use several GIN blocks to map from $\mathbb{R}^n \to \mathbb{R}^n$. Recalling that each GIN block is an injection (since part of it is an identity map, one can not map two different inputs to the same output), it follows that the composition of several blocks remains an injection. While we use an extension of GIN (64) to implement the injection $f$ here, conceivably other implementations should be possible, e.g., multi-layer perceptron with increasing number of nodes from earlier to later layers. The efficiency of the different implementations will need to be evaluated in future.

## 2.2 Inference algorithm

The inference procedure is a modification of VAE (37). Our algorithm simultaneously learns the deep generative model and the approximate posterior $q(\mathbf{z}|\mathbf{x}, \mathbf{u})$ of true posterior $p(\mathbf{z}|\mathbf{x}, \mathbf{u})$ by maximizing $\mathcal{L}(\boldsymbol{\theta}, \boldsymbol{\phi})$, which is the evidence lower bound (ELBO) of $p(\mathbf{x}|\mathbf{u})$,

$$\log p(\mathbf{x}|\mathbf{u}) \geq \mathcal{L}(\boldsymbol{\theta}, \boldsymbol{\phi}) = \mathbb{E}_{q(\mathbf{z}|\mathbf{x}, \mathbf{u})} \left[\log\left(p(\mathbf{x}, \mathbf{z}|\mathbf{u})\right) - \log\left(q(\mathbf{z}|\mathbf{x}, \mathbf{u})\right)\right]. \tag{5}$$

Similar to (31), we decompose the approximate posterior as

$$q(\mathbf{z}|\mathbf{x}, \mathbf{u}) \propto q_{\boldsymbol{\phi}}(\mathbf{z}|\mathbf{x})p_{\mathbf{T}, \boldsymbol{\lambda}}(\mathbf{z}|\mathbf{u}), \tag{6}$$

where $q_{\boldsymbol{\phi}}(\mathbf{z}|\mathbf{x})$ is assumed to be conditionally independent exponential family distribution, i.e. $q_{\boldsymbol{\phi}}(\mathbf{z}|\mathbf{x}) = \prod_{i=1}^{m} q(\mathbf{z}_i|\mathbf{x})$, and is parameterized by neural network. $p_{\mathbf{T}, \boldsymbol{\lambda}}(\mathbf{z}|\mathbf{u})$ is defined in equation 2.

We modeled both $q_{\boldsymbol{\phi}}(\mathbf{z}|\mathbf{x}), p_{\mathbf{T}, \boldsymbol{\lambda}}(\mathbf{z}|\mathbf{u})$ as independent Gaussian distribution, used the same network architecture (see SI for details) as well as Adam optimizer (36) with learning rate equal to $5 \times 10^{-4}$, and other values were set to the recommendation values for all the experiments in this paper.

**Inferring the latent** After learning $q(\mathbf{z}|\mathbf{x}, \mathbf{u})$, the latent from pi-VAE model can be inferred by calculating the posterior mean. It is also of interest to infer the latent without using the label prior $p_{\mathbf{T}, \boldsymbol{\lambda}}(\mathbf{z}|\mathbf{u})$, which could be done by calculating the posterior mean estimate of $q_{\boldsymbol{\phi}}(\mathbf{z}|\mathbf{x})$ instead.

**Decoding the label** Because pi-VAE defines an encoding model on the label, one can examine how well the label could be decoded from the neural activity, which also provides a way to check the validity of the model. Under our model formulation, decoding could be done by Bayesian rule and Monte Carlo sampling: $p(\mathbf{u}|\mathbf{x}) \propto \int p(\mathbf{x}|\mathbf{z}, \mathbf{u})p(\mathbf{z}|\mathbf{u})p(\mathbf{u})d\mathbf{z}$, where the integration on right hand side can be computed through randomly sampling $p(\mathbf{z}|\mathbf{u})$. We assume a uniform prior on $\mathbf{u}$.

## 3 Results

### 3.1 Validation using synthetic data

We validated pi-VAE using synthetic data generated from models with continuous or discrete labels.

**Discrete label** We generated 2-dimensional latent samples $\mathbf{z}$ from a five clusters Gaussian mixture model, similar to (35) (see Fig. 2a). The mean of each cluster was chosen independently from a uniform distribution on $[-5, 5]$ and variance from a uniform distribution on $[0.5, 3]$. These latent samples were then mapped to the mean firing rate of 100-dimensional Poisson observations through a RealNVP network (details in SI). Example results are shown in Fig. 2a-d.

**Continuous label** We generated $\mathbf{u}$ from a uniform distribution on $[0, 2\pi]$, and latent samples $\mathbf{z}$ as a 2-dimensional independent Gaussian distribution with mean being $(\mathbf{u}, 2\sin\mathbf{u})$, and variance being

$(0.6 - 0.3|\sin \mathbf{u}|, 0.3|\sin \mathbf{u}|)$. Observations were generated in the same way as simulation of discrete label. Example results are shown in Fig. 2e-h.

Based on these and other numerical experiments, we found that in general pi-VAE could reliably uncover latent structure similar to ground truth for both discrete and continuous labels, while VAE often leads to more distorted latent. Note that our VAE implementation is similar to pi-VAE except that no label prior is used (*e.g.*, Poisson observation noise is still assumed). We also found pi-VAE without label prior (during the inference) still led to reasonably good recovery of the latent (Fig. 2c,g), suggesting that incorporating label prior could help with learning a better model, not just inference.

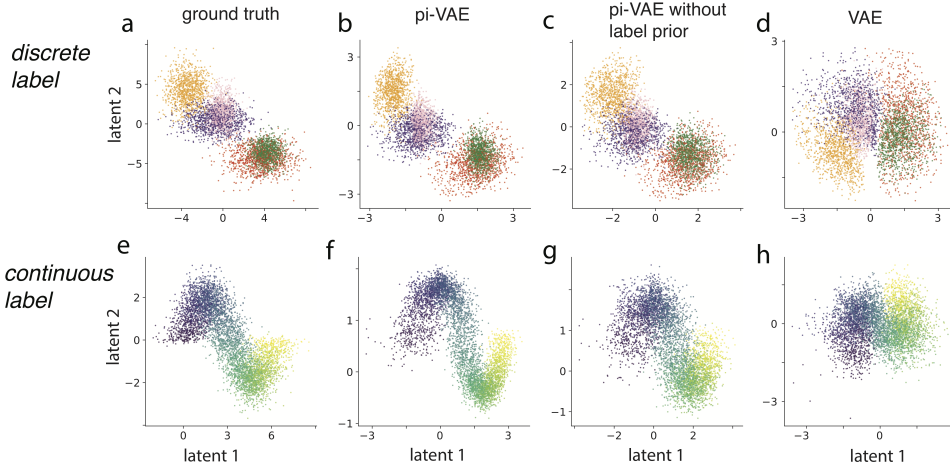

Figure 2: Example numerical experiments, suggesting that pi-VAE, but not VAE, could identify latent structure. (a) True latent variables, simulated based on discrete label, (b) mean of the latent posterior $q(\mathbf{z}|\mathbf{x}, \mathbf{u})$ estimated from pi-VAE, (c) mean of $q(\mathbf{z}|\mathbf{x})$ from pi-VAE, (d) mean of the latent posterior from VAE, that is, the Bayesian estimate *inferred* without the label prior. (e-h) similar to (a-d), but for a simulation based on continuous label.

## 3.2 Applications to neural population data

We have applied pi-VAE to analyze two electrophysiology datasets, each has more than 100 simultaneously recorded neurons when the animals were performing behavioral tasks. In these real data applications the ground truth is unknown and the assumptions required by identifiability may be violated (64), making it difficult to assess identifiability directly. Our rationale is that, assuming the ground truth is structured, models with better identifiability would still lead to more interpretable latent representation. Encouragingly, examination of the latent space extracted from these datasets indeed suggest that pi-VAE could extract interpretable and meaningful latent structure.

### 3.2.1 Monkey reaching data

We first applied our method to a previously published monkey reaching datasets (see (21), kindly shared by the authors). In these experiments, Monkey C was performing a reaching task with 8 different directions, while neural activities in areas M1 and PMd were simultaneously recorded (for details, see (21)). We analyzed two sessions, and obtained similar results. We will focus on Session 1 here, and detailed results from Session 2 can be found in SI.

For each direction, there are $\sim 25$ trials/repeats (see Fig. 3a). We analyzed 192 neurons from PMd area, and focused on the reaching period from go cue (defined as $t = 0$) to the end, which typically lasts for $\sim 1$ second. We binned the ensemble spike activities into 50ms bins. We used the spike activities as observation $\mathbf{x}$, and the reaching direction as the discrete labels $\mathbf{u}$. We randomly split the dataset into 24 batches, where each batch contains at least one trial for each direction. We randomly split them into training, validation and test data $(20, 2, 2$ batches$)$. We fit 4-dimensional latent models to the data based on pi-VAE and VAE.

**Goodness of fit** We first assessed the goodness of fit by examining the root-mean-square error (RMSE) of the PSTH based on the prediction of each model (see Fig. 3a for an example neuron).

Fig. 3b,c show that pi-VAE leads to the smallest RMSE of firing rate in most neurons, followed by VAE, then tuning curve model. Next, we computed the log marginal likelihood $p(\mathbf{x})$ on the held-out test data by randomly sampling both $p(\mathbf{z}|\mathbf{u})$ and $p(\mathbf{u})$. We found that pi-VAE leads to larger mean marginal log-likelihood than VAE and tuning curve model ($-123, -123.4, -127.6$ respectively, t-test $p < 10^{-6}$). These results suggest that pi-VAE provides the best fit to the data among the alternatives.

**Decoding reaching direction** We wondered whether pi-VAE also provides a better encoding model of the reaching direction. We examined how well pi-VAE could decode reaching direction, and compare the performance to a traditional method based on direction tuning curves. On held-out test data, pi-VAE achieved an average single-frame (50ms) decoding accuracy of $61\%$, better than $47\%$ from tuning curve model. Examination of the time course of the decoding performance (Fig. 3d) reveals that pi-VAE achieves $60\%$ during the first few frames before initiation of reaching, while tuning curve model is much worse during this period. However, when reaching speed reaches its maximum (around 0.5s, Fig. 3e), both models achieve almost perfect performance (Fig. 3d). These observations tentatively suggest that information about the reaching direction may be encoded in different format during different phases of reaching in this task.

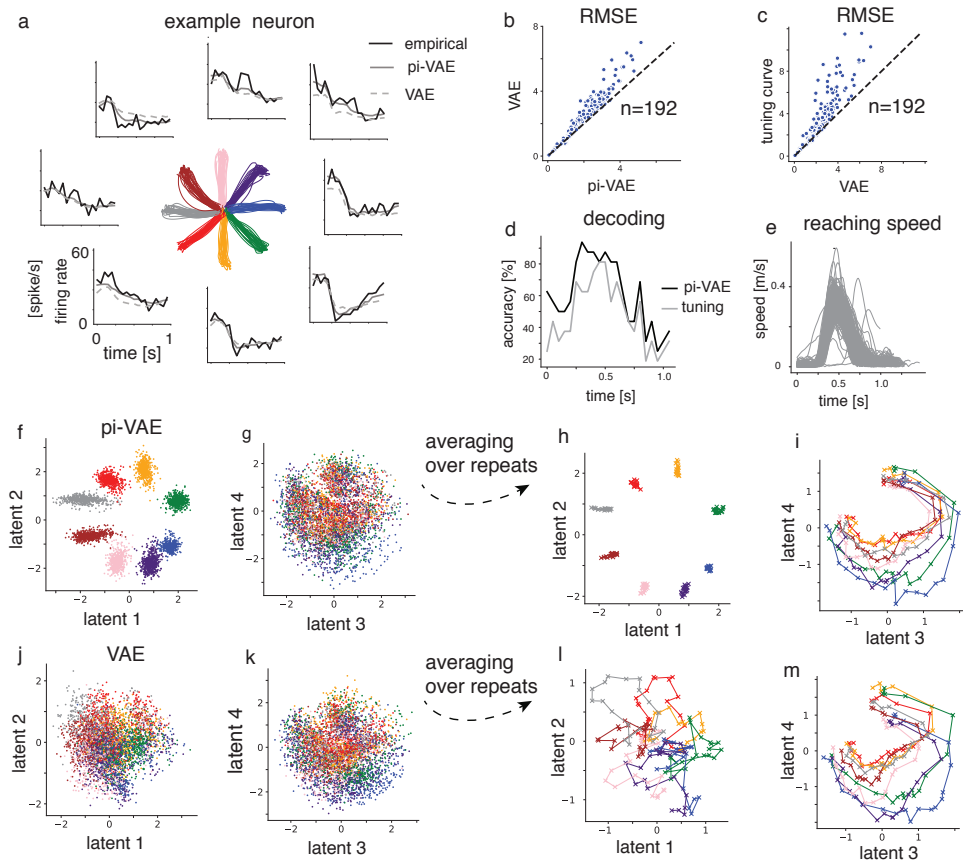

Figure 3: Monkey reaching data. (a) Reaching trajectories for $8$ directions labeled by colors. The empirical firing rate (PSTH, black solid line), fitted rate by pi-VAE (gray solid line) and VAE (gray dashed line) for an example neuron. (b,c) Scatter plots of RMSE of fitted rate ($n = 192$ neurons) for comparing pi-VAE and VAE, as well as VAE and tuning curve. (d) Decoding accuracy as function of time on test data by pi-VAE and tuning curve model. (e) The reaching speed of the macaque for each trial. (f,g) Inferred latent based on pi-VAE, i.e.,mean of $q(\mathbf{z}|\mathbf{x}, \mathbf{u})$. (h,i) Inferred latent from pi-VAE averaged over repeats from the same reaching direction. (j,k,i,m) Similar to (f,g,h,i) for VAE. Notice the striking difference between (f) and (j).

**Structure of the latent** We found that the latent variables estimated by pi-VAE exhibit clear structures. To start, the $8$ reaching directions are well separated in the subspace defined by the first two latent dimensions (Fig. 3f,h). Strikingly, the geometrical structure of the inferred latent resembles the

geometry of the reaching directions. In contrast, the third and fourth latent dimensions captures the evolution of the trajectories over time, and they are only weakly informative about reaching directions (Fig. 3g,i). Thus, the axes of the extracted latent space are easily interpretable. They provide information about how reaching direction is represented, and how neural dynamics evolve during reaching behavior. Notably, these axes were extracted automatically from pi-VAE, and no additional factor analysis techniques were applied to identify salient latent axes. Strikingly, the latent variables extracted from Session 2 show very similar structure (see SI Fig. S1).

In comparison, VAE extracts a much more entangled latent representation (Fig. 3j-m). It appears that information about reaching direction displays in a twisted fashion, and mixes with the temporal evolution of the trajectories. Note that these differences between the latent structure obtained from two methods is not simply due to the label prior. The inferred latent from pi-VAE without the label prior (*i.e.*, posterior mean of $q(\mathbf{z}|\mathbf{x})$) shows similar though a bit more diffuse latent structure, which is expected due to the observation noise (see SI Fig. S2).

The nature of the neural code during primate reaching behavior is currently under heavy debate (24; 23; 60; 65; 13; 61; 40; 21). While earlier proposals emphasized the encoding of task relevant variables (24; 23; 60; 53), some of the more recent studies instead highlighted the importance of neural dynamics (13; 61; 2; 32). As shown above, pi-VAE discovers latent space that exhibits striking spatial (*i.e.*, reaching direction) *and* temporal (*i.e.*, neural dynamics) structure that are separately encoded in different sub-spaces. These preliminary results may provide a way to reconcile the two prominent hypotheses (24; 61), as evidence for both hypotheses are now revealed in the same model based on the same datasets. It would be important to apply our methods to larger datasets from multiple monkeys to examine the consistency of these effects in future.

### 3.2.2  Rat hippocampal CA1 data

We next applied pi-VAE to analyze a public rat's hippocampus dataset (27; 26)[1]. In this experiment, a rat ran on a 1.6m linear track with rewards at both ends (L&R) (Fig. 4a), while neural activity in the hippocampal CA1 area was recorded ($n = 120$, putative pyramidal neurons). We focused on the data when the rat was running on the track (Fig. 4a) and binned the ensemble spike activities into 25ms bins. We defined the rat running from one end of the track to the other end as one lap, resulting in 84 laps. We randomly split them into training, validation and test data (68, 8, 8 laps). We defined rat's position and running directions as continuous labels $\mathbf{u}$. We fit 2-dimensional latent models to the data for both pi-VAE and VAE.

**Goodness of fit and decoding performance** We found that pi-VAE again outperformed alternatives in having the lowest mean log marginal likelihood $-17.7$ (VAE, $-17.9$; tuning curve, $-18.2$; paired t-test, $p < 10^{-6}$). Furthermore, we decoded the animal's location on the tracking based on pi-VAE model and tuning curve model. On test data, pi-VAE achieves median absolute decoding error (MAE) of 12cm (time window = 25ms), while the tuning curve (traditional "place field" (49)) model achieves a MAE of 15cm. This indicates that for the simple purpose of constructing an effective encoding model of the animal's position on the track, pi-VAE outperforms the traditional place field model (73).

**Structure of the latent** Fig. 4b shows the latent space estimated by the pi-VAE, which exhibits overtly interpretable geometry: the collection of inferred latent states for R-to-L (blue) or L-to-R (red) running direction each forms band-like sub-manifold, and both are roughly in parallel with the second latent dimension (Fig. 4b). The split into two sub-manifolds is consistent with the observation that place fields of CA1 neurons often have firing fields that are uncorrelated between the two travel directions ("directional" firing) (3; 14; 48). To further quantify the geometrical relation between two sub-manifolds, we calculated the distance for pairs of points from the two branches(Fig. 4b). This quantification for every possible pair (after binning the position into 16cm bins) is plotted in Fig. 4c. We found that the manifold geometry across the two directions respects the geometry of the track, in the sense that smaller physical distance on the track leads to smaller latent distance. This is likely due to that a subset of place cells have non-directional (purely spatial) place fields (3; 14; 48; 33). Importantly, our method gives a quantitative population level characterization of the consequence of having both directional and non-directional place cells. pi-VAE without label prior shows similar but more diffuse latent structure (see SI Fig. S3). In contrast, VAE results in a tangled latent representation, with the geometry not reflecting physical distance on the track (Fig. 4d,e; also notice that striking difference between Fig. 4b and Fig. 4d).

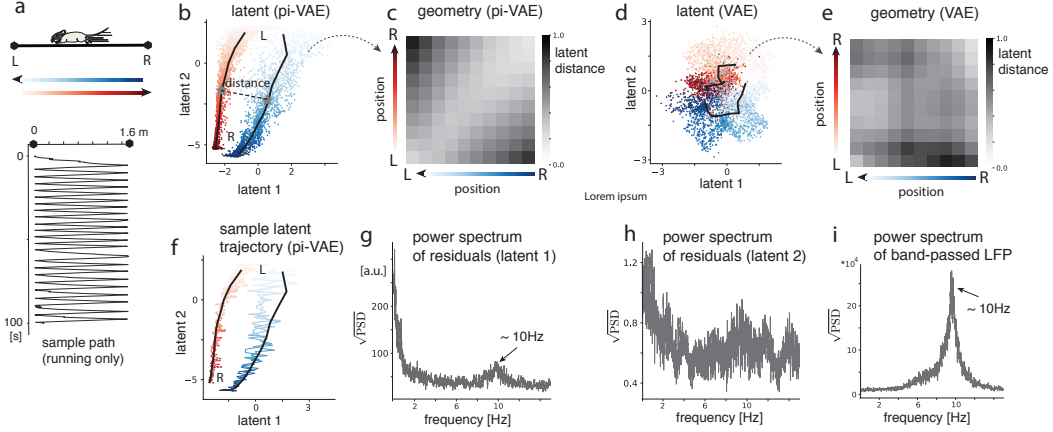

Figure 4: Results on hippocampus CA1 data. (a) Linear track and sample running path. The two ends are labeled as L&R. Two directions are color-coded by red and blue, and positions are coded by color saturation. (b) Inferred latent from pi-VAE. Black lines represent the mean of the latent states corresponding to position on the track for two directions. The distance between pairs of points from the two black lines is computed to quantify the latent geometry. An example pair of points are indicated using grey stars. The normalized distance for all possible pairs of points is shown in panel (c). (d,e) are defined similar to (b,c) for the VAE. Notice the striking difference between (b) and (d). (f) Two sample latent trajectories of the pi-VAE. (g,h) The power spectrum (PSD) of the residuals of the latent, given by the mean of $q(\mathbf{z}|\mathbf{x}, \mathbf{u})$ minus the mean of $p(\mathbf{z}|\mathbf{u})$. (i) PSD of the band-passed LFP in the range of 5-11 Hz.

To investigate whether the latent model could yield additional scientific insight, we next examined the temporal structure of the latent. Fig. 4f plots sample trajectories, from which we observed that temporal fluctuation mainly goes along the first latent dimension, and the suggestion of rhythmic structure. We subtracted the mean of prior $p(\mathbf{z}|\mathbf{u})$ from the mean of posterior $q(\mathbf{z}|\mathbf{x}, \mathbf{u})$ to obtain the residual fluctuations. Examination of the power spectrum density (PSD) along each dimension of the residuals led to two observations: i) the temporal fluctuation is indeed concentrated on the first latent dimension, as indicated by the magnitude of the PSD; ii) the first, but not the second, dimension exhibits a striking peak at $\sim 10$Hz. We reasoned that the second observation might be related to $\theta$-oscillation in the local circuit, which is known to modulate the firing of CA1 neurons (62; 20; 55; 33). We thus examined the simultaneously recorded local field potential (LFP) data during running. Indeed, we found that the $\theta$ peaked at $\sim 10$Hz for this rat. Interestingly, the 10Hz $\theta$-oscillation is faster than the typically reported 8 Hz average frequency (69; 9), yet is consistent with the latent structure extracted from pi-VAE.

Overall, pi-VAE extracts latent space that is clearly interpretable, with one dimension encoding position information, and the other dimension capturing temporal organization which is likely related to $\theta$-rhythm. The observation that the position encoding and the rhythmic-like fluctuation are roughly orthogonal is particularly interesting, and is consistent with previous results from the single cell analysis suggesting that theta phase and place fields may encode independent information (30). Additional investigations will be needed to test these hypotheses in greater depth.

### 3.3 Comparison to alternative methods

We further tested several alternative methods on the monkey reaching data, including both linear methods (demixed PCA (38)) and nonlinear methods (UMAP (47), PfLDS (22), and LFADS (52)) (see Fig. 5). Overall we found that, while the extracted latent structures from these methods exhibited interesting characteristics, none of them resulted in fully disentangled latents. Furthermore, none of them appeared to recover the geometry of the physical reaching targets. (More analysis of the hippocampus data can be found in SI Sec. E)

To start, supervised UMAP recovered latents corresponding to different directions as different clusters, but without clear representations of temporal dynamics(see Fig. 5a). Furthermore, LFADS (52) and

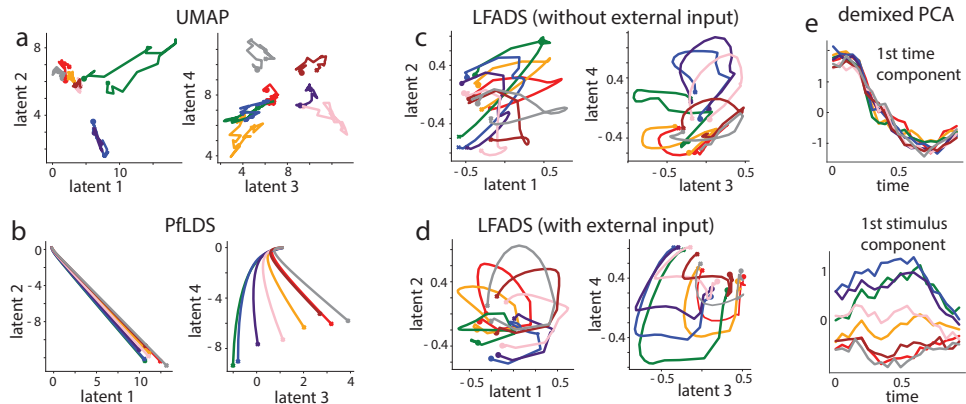

Figure 5: Results from alternative methods based on monkey reaching data. a) UMAP (47). b) PfLDS (22). c,d) LFADS (52), without and with reaching direction as an external input. e) demixed PCA (38) with the first time and stimulus component plotted. Color-coded, averaged latent trajectories corresponding to each reaching direction was plotted for each method. The filled dot and cross represent starting and ending of the trial.

PfLDS (22) both led to smooth trajectories. Although the trajectories for different directions were separated in the 4-dimensional space, directions and temporal dynamics were entangled so that it was difficult to interpret each individual latent dimension (Fig. 5b,c,d). Demixed PCA (38) with both time and directions as labels still entangled time and directions (stimulus components change with time) to some extent (Fig. 5e). A few methodological considerations are worth mentioning here. First, LFADS can take task variables as external inputs to the model RNN. We thus tried LFADS with or without reaching direction as external inputs (Fig. 5c,d). Second, demixed PCA only deals with discrete task variables each with the same number of trials and each trial with the same length, and could not recover additional latent fluctuations as our method. Third, UMAP can incorporate label information for supervised learning, and we used the reaching directions as labels (Fig. 5c,d) to make a more fair comparison. However, we found that it did not recover temporal dynamics.

## 4   Discussion

We have presented a new model framework for analyzing neural population data by integrating ingredients from latent-based and regression-based approaches. Our model pi-VAE, while being expressive and nonlinear, is constrained by additional dependence on task variables. pi-VAE generalizes recent work on identifiable VAE (35; 41; 64) to deal with spike train data. Although pi-VAE yields promising preliminary insights into the neural codes during a rat navigation task and macaque reaching task, we should emphasize that more systematic investigations based on larger datasets across different subjects will be needed to further elaborate these results.

Our method is motivated by leveraging the strength of regression-based methods and latent-based models to increase the identifiability and interpretability, a direction received little attention previously. To do so, we took advantage of the "label prior" to model the impact of task variables on neural activities along with the influence of the latent states. One potential concern is that, when incorporating too many labels, there may not be enough data to fit the model. Several previous methods exploited temporal smoothness priors to de-noise the data, which were implemented via Gaussian process (10; 74; 71), linear (44; 7; 22) or nonlinear dynamical systems (52; 17). Although not pursued here, adding temporal smoothness priors into pi-VAE may increase the data efficiency and further improve the performance of the model. It is also worth mentioning that although we have focused on the spike train data, our method may be modified to deal with the calcium imaging data incorporating noise models that is more appropriate to the deconvolved calcium traces (68). Last but not least, while the current study mainly concerns the neuroscience applications of pi-VAE, some of the technical advances made here may be of interest to the machine learning community as well.

**Acknowledgements** We thank Matthew G. Perich and Lee E. Miller for sharing the monkey reaching data. We thank Kenneth Kay, Yuanjun Gao and Liam Paninski for helpful comments and discussions, as well as three anonymous reviewers for their feedback which helped improving the paper. Xue-Xin Wei is supported by the startup funds provided by UT Austin.

## 5   Broader Impact

In this paper, we develop a new analysis method for analyzing neural population data. This method can be used to extract and identify the underlying critical structure underlying the neural activity recorded simultaneously from many neurons in the brain. It can also be used to construct improved response models of how various kinds of task variables are encoded in the brain. Our approach is of broad applicability to the neural population recording under a variety of scenarios. It may also inspire future work in neural data analysis. Progress in this area will facilitate both basic science research about the brain as well as clinical applications, such as invasive and non-invasive brain-machine interfaces.

Our method is developed based on recent advances in a class of deep generative models, i.e., identifiable variational auto-encoder. It may be of interest to the machine learning community working on deep generative models. Our research extends previous works on identifiable variational auto-encoder to a different setup. Our preliminary promising results may provide useful insights into the further development of more identifiable deep generative models, which in the longer term may help establish more interpretable and robust AI technology, and help understand the limitations of these technologies.

## Footnotes

[1] http://crcns.org/data-sets/hc/hc-11

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
