[Supplementary Material]

# Supplementary Information (SI)

## A    Proof of identifiability for pi-VAE

**Theorem 1.** *Assume that we observe data sampled from pi-VAE model defined according to equation 1,2 with Poisson noise and parameters $\boldsymbol{\theta} = (\mathbf{f}, \mathbf{T}, \boldsymbol{\lambda})$. Assume the following holds:*

*i) The firing rate function $\mathbf{f}$ in equation 1 is injective.*

*ii) The sufficient statistics $T_{i,j}$ in 2 are differentiable almost everywhere, and their derivatives $T'_{i,j}$ are nonzero almost everywhere for $1 \leq i \leq m, 1 \leq j \leq k$.*

*iii) There exists $mk + 1$ distinct points $\mathbf{u}^0, \cdots, \mathbf{u}^{mk+1}$ such that the matrix*

$$L = \left( \boldsymbol{\lambda}(\mathbf{u}^1) - \boldsymbol{\lambda}(\mathbf{u}^0), \cdots, \boldsymbol{\lambda}(\mathbf{u}^{mk}) - \boldsymbol{\lambda}(\mathbf{u}^0) \right)$$

*of size $mk \times mk$ is invertible, then the pi-VAE model is identifiable up to $\sim$.*

*Proof.* (35) has proved that the Bernoulli observation model is identifiable under the same set of assumptions. For Poisson observations with mean firing rate as $\lambda$, we can transform it to Bernoulli observations with parameter $p = 1 - \exp(-\lambda)$ by keeping the zeros and treating the positive values as ones. Because the Bernoulli model is identifiable, the Poisson model is also identifiable. $\qquad\square$

## B    Network architecture

For both the generative models in pi-VAE and VAE, we used the following strategy to parameterize $\mathbf{f}(\cdot)$ which maps the $m$-dimensional latent $\mathbf{z}$ to the mean firing rate of $n$-dimensional Poisson observations. We first mapped the $\mathbf{z}_{1:m}$ to the concatenation of $\mathbf{z}_{1:m}$ and $t(\mathbf{z}_{1:m})$, where $t(\cdot) : \mathbb{R}^m \to \mathbb{R}^{n-m}$ is parameterized by a feed-forward neural network with a linear output and 2 hidden layers, each containing $\lfloor n/4 \rfloor$ nodes with ReLU activation function. Then we applied two GIN blocks. Same as (64), we defined the affine coupling function as the concatenation of the scale function $s$ and the translation function $t$, computed together for efficiency, applied two affine coupling functions per GIN block, and randomly permuted the input before passing it through each GIN block. We defined both $s, t$ in GIN block as mapping: $\mathbb{R}^{\lfloor n/2 \rfloor} \to \mathbb{R}^{n-\lfloor n/2 \rfloor}$. The scale function $s$ is passed through a clamping function $0.1\tanh(s)$, which limits the output to the range $(-0.1, 0.1)$. For affine coupling function, we have a linear output layer with and 2 hidden layers, each containing $\lfloor n/4 \rfloor$ nodes with ReLU activation function.

We modeled the prior $p_{\mathbf{T},\boldsymbol{\lambda}}(\mathbf{z}|\mathbf{u})$ in pi-VAE as independent Gaussian distribution. The natural parameters $\lambda_{i,j}$ are the Gaussian means and variances. For discrete $\mathbf{u}$, we used different values of the mean and variance for different labels. For continuous $\mathbf{u}$, we parameterized the mean and spectrum decomposition of variance together by a feed-forward neural network with a linear output layer and 2 hidden layers, each containing 20 nodes with tanh activation function (the mean and variance share the 2 hidden layers). For the cases of mixed discrete and continuous labels $\mathbf{u}$, we encoded the discrete labels with a one-hot vector, and mapped it together with the continuous components to the mean and spectrum decomposition of variance using feed-forward neural network as described in the continuous case.

For the recognition model in pi-VAE and VAE, we used $q_\phi(\mathbf{z}|\mathbf{x})$ as independent Gaussian distribution, and parameterized the mean and the spectrum decomposition of the variance separately using feed-forward neural network with a linear output layer and 2 hidden layers, each containing 60 nodes with tanh activation function.

Code implementing the algorithms is available at https://github.com/zhd96/pi-vae.

## C    Synthetic data simulations

To generate firing rate of the Poisson process from simulated latent $\mathbf{z}$, we first padded $\mathbf{z}$ with $n$-$m$ zeros, then applied 4 RealNVP blocks, each containing 2 affine coupling functions with the same structure as defined in section B except that $s$ does not need to have sum equal to 0 here, and we used $\lfloor n/2 \rfloor$ nodes for each hidden layer.

For discrete label simulation shown in Fig. 2a-d, we simulated $10^4$ observations, and split them into training, validation, test data ($80\%, 10\%, 10\%$ respectively). We set the batch size to be 200 during training, and trained for 600 epochs. For the continuous label simulation shown in Fig. 2e-h, we simulated $1.5 \times 10^4$ observations. The training-validation-test split is the same as discrete label simulation. We set batch size as 300, and trained for 1000 epochs.

## D   Monkey reaching data: session 2

For each reaching direction, there are $\sim 35$ trials (see Fig. S1a). We analyzed 211 neurons from PMd area, and focused on the reaching period from go cue (defined as $t = 0$) to the end, which typically last for $\sim 1$ second. We binned the ensemble spike activities into 50ms bins. We randomly split the dataset into 34 batches, where each batch contains at least one trial from each direction. We randomly took 28 batches as training data, 3 batches as validation data and 3 batches as test data. Similar to Session 1, We fit 4-dimensional latent models to the data based on pi-VAE and VAE respectively. Results are shown in Fig. S1.

## E   Alternative methods

### E.1   Monkey reaching data

For supervised UMAP[2], we set the reaching directions as labels, and embedded the high dimensional spike count data into a 4-dimensional latent space. Other parameters were set to be the default values. For PfLDS, we implemented the algorithm on our own using the same neural network architecture as in (22) (the original code provided by the authors of that paper depends on Python Theano library, which has not been maintained for a while). We assumed a 4-dimensional latent space and Poisson observation model. We set the learning rate as $2.5 \times 10^{-4}$ and trained for 1500 epochs. Each batch consisted of a single trial. The training, validation and test sets had $184, 16, 17$ trials respectively. For LFADS[3], we assumed a 4-dimensional latent space along with a Poisson observation model. We applied two versions of the model to the data, with the reaching direction as an additional input and without this input. Other parameters were set to be the default values. We pre-processed the data by discarding all the trials less than 1 second and trimming longer trials to make them 1 second long. Each batch consisted of a single trial. The training, validation and test sets had $177, 16, 16$ trials respectively. For demixed PCA[4], we pre-processed the data to make each reaching direction had the same number of trials and each trial had the same length (*i.e.*,1 second). We took time and stimulus as labels. We used 2 sets of components, each containing time, stimulus, as well as time and stimulus mixing components. Other parameters (eg. regularizer) were set to be the default values.

### E.2   Hippocampus data

For supervised UMAP, we used 2-dimensional latent variables model with rat's locations as labels. Other parameters were set as default. For PCA, we used two principal components. For "PCA after LDA", we first applied LDA with rat's running directions as response and neural activities as predictors, and identified the 1-dimensional linear boundary which could separate the neural activities of two directions most. Then we projected the neural activities on this boundary using linear regression, then applied PCA with 2 principal components on the residuals. We found that the resulting latents were all less interpretable than pi-VAE (Fig. S4), with no dimension directly representing the rat's location. Also the rhythmic-like fluctuations spanned across dimensions, rather than concentrated in one dimension (not shown).

Figure S1: Results on Macaque reaching data (Session 2). These results are similar to those obtained from Session 1 as reported in the main text. (a) The macaque's reaching trajectories for 8 directions labeled by different colors. (b) The reaching speed of the macaque for each trial. (c,d) Scatter plots of RMSE of fitted rate ($n = 211$ neurons) for comparing pi-VAE and VAE, as well as VAE and tuning curve. (e) Decoding accuracy as function of time on test data by pi-VAE and tuning curve model. (f,g) Inferred latent based on pi-VAE, i.e.,mean of $q(\mathbf{z}|\mathbf{x}, \mathbf{u})$. (h,i) Inferred latent from pi-VAE averaged over repeats from the same reaching direction. (j,k) Mean of $q(\mathbf{z}|\mathbf{x})$ from pi-VAE. (l,m) Mean of $q(\mathbf{z}|\mathbf{x})$ by pi-VAE averaging over repeats from the same reaching direction. (n-q) Similar to (f-i) for VAE.

Figure S2: Related to Fig. 3, on reaching data. Inferred latent without label prior using pi-VAE still are still highly structured and interpretable. The first two dimensions carry information about the reaching direction, while the third and fourth dimension mainly captures the dynamics over the time course of a trial. (a,b) Mean of $q(\mathbf{z}|\mathbf{x})$ from pi-VAE. (c,d) Mean of $q(\mathbf{z}|\mathbf{x})$ by pi-VAE averaging over repeats from the same reaching direction.

Figure S3: Related to Fig. 4. Results on Hippocampus CA1 data. Inferred latents without the label prior using pi-VAE still exhibit clear structure, with the latent geometry respecting the geometry of the track. (a) Mean of $q(\mathbf{z}|\mathbf{x})$ from pi-VAE. Two directions are color-coded by red and blue, and positions are coded by color saturation. Black lines represent the mean of the latent states corresponding to position on the track for two directions. (b) The distance between pairs of points from the two black lines is computed to quantify the latent geometry.

Figure S4: Hippocampus data: results from several alternative methods. a) UMAP. b) PCA. c) PCA after Linear Discriminant analysis (LDA). Notice that these methods recovered more entangled representation compared to pi-VAE.

## Footnotes

[2]https://umap-learn.readthedocs.io/en/latest/

[3]https://github.com/lfads/models

[4]https://github.com/machenslab/dPCA