[Reviews · NeurIPS 2020]

Review 1

Summary and Contributions: The paper proposes a slight modification of the recently proposed iVAE framework to deal with spike data. The main contribution is showing how this leads to a major improvement on VAE in the analysis of multielectrode data.

Strengths: The spiking data results are very well done and show great potential for the proposed method.

Weaknesses: The theory is a meaningful extension of the iVAE framework, but its novelty seems to be more limited than what is claimed (see below).

Correctness: I think the results are correct, but I have some problem with the novelty claims, see below.

Clarity: It is very well written. Just a minor detail: The references are often given as very long lists, so it is difficult for the reader to understand which of the references they should read to find the right source.

Relation to Prior Work: In general yes, but: on line 69 as well as in the conclusion, the novelty of the model is described as enabling "Poisson noise, continuous labels, and larger output dimension than input dimension". I think the two latter ones are in the iVAE paper already, in particular in its Theorem 1. (Poisson noise is definitely new.) So, the novelty of the method is a bit weaker than claimed, in my opinion.

Reproducibility: Yes

Additional Feedback: Two requests for clarification: I'm a bit confused by Fig 2c which says "without label prior". Does this means the label prior was not used in the inference or the learning, or neither? I guess only in inference, in which case the legend in Fig 2 would rather be "inferred without label prior". Nor is it quite clear to me if your VAE, when used for comparison, used Poisson noise. A comment: If I understand correctly, your main claim claim to fame is the quite striking difference between Fig 3f and 3j on the one hand, and the equally big difference between 4b and 4d on the other. If this is so, you could emphasize that point, since there are so many plots in the paper that this may get lost.


Review 2

Summary and Contributions: The authors adapt modern deep learning methods to develop a latent variable model for exploratory analyses of neural data. The modeling innovations appear minor to me, but the authors describe relatively thorough analyses of two experimental datasets, with more details than a typical NeurIPS submission. The results are somewhat compelling, though it is unclear whether this type of approach will lead to new scientific insights that couldn't be found with simpler methods.

Strengths: - Nice in depth analysis of two experimental datsets - Imports cutting edge methodologies from deep learning - Emphasizes interpretability as a key desiderata / advantage of the method

Weaknesses: - A large number of (nonlinear) latent variable models have already been proposed for neural data analysis. Not enough comparisons are made to these existing methods. - Though the two experimental datasets are analyzed in detail, it is not clear whether crafting better nonlinear latent variable models will lead to different scientific insights. These kinds of models may be better suited for tasks with quantitative benchmarks (e.g. better spike sorting, segmenting calcium imaging datasets, etc.) than qualitative exploratory data analysis.

Correctness: The claims and methods appear correct to me.

Clarity: Yes.

Relation to Prior Work: One of the weaknesses of the paper is that it narrowly compares the performance of the new method pi-VAE to a traditional VAE, but does not consider other relevant baselines that are cited as prior work. Models that spring to mind are those of Gao et al. (2016) and Pandarinath et al. (2018). Also the paper "Deep Random Splines for Point Process Intensity Estimation of Neural Population Data" that appeared at NeurIPS last winter would be a good citation to add. One line of research that is not well-cited or considered would be "supervised dimensionality reduction" (also "semi-supervised"). I know these methods handle the case of discrete covariates (the "u" variable) quite well, and probably also work well for continuous "u". The simplest method would be linear discriminant analysis (LDA). A more complex one would be UMAP (McInnes et al., 2018) which has been extended to the supervised case. In the neuroscience literature, demixed PCA (Kobak et al., 2016) seems like a very important paper that should be compared with and cited...

Reproducibility: Yes

Additional Feedback: On Reproducibility: - I think the basic methodology could be replicated, but it would have been nice to include code as a supplementary material. I hope the authors can assure me that the code will be documented and made available upon publication. On The Motor Cortex Dataset: - The VAE and pi-VAE seem to perform similarity in panel i and panel m - The better performance of pi-VAE in panel h vs l is likely due to the input variable "u" which forces different latent represenations (update: after writing this, I noticed that this is indeed the case based on supplementary figure S1; though pi-VAE is still slightly better). This is fine, but perhaps makes the result unsurprising -- wouldn't other supervised methods (e.g. demixed PCA, or supervised UMAP) achieve similar results? On The Hippocampal Dataset: - For fig 4B, I think that linear discriminant analysis (LDA) would be sufficient to get you separation between the two running directions --- i.e. this would recover "latent 1". If you then project out this dimension and found the top PC in the remaining data, I conjecture this would give you something similar to "latent 2". This would be a purely linear dimensionality reduction / latent variable model, and it probably would not capture the theta-rhythmicity. Nonetheless, I would have really liked to see analyses like this to understand what these kinds of nonlinear latent variable models buy us. ---- RESPONSE TO AUTHOR FEEDBACK ---- Overall, I'm very impressed that the authors were able to include a number of new baselines in their response. After much agonizing between choosing a 5 (marginal reject) and a 6 (marginal accept), I've decided to give the benefit of the doubt and settled on a score of 6. I have two main reservations about the work: 1) In general, I think nonlinear unsupervised learning methods (pfLDS, LFADS, etc.) have somewhat limited utility. Despite being around for several years, I haven't yet seen any novel scientific insights arise from this general line of work. That being said, better tools for reverse-engineering and understanding deep network models may help overcome this in the future. This is still an area worthy of research, but it is a bit speculative, in my view. 2) Given my feelings on point 1 above, I think it is really important for papers like this to compare to simple baselines (mostly, to linear methods like LDA, PCA, demixed PCA). I feel this was severely lacking in the initial paper. Though the author response begins to address my concern, I am conflicted about how well I can evaluate these new comparisons given the restricted space of the 1 page response. For example, I am somewhat surprised that supervised UMAP does not give good results and I worry that a more careful tuning of the hyperparameters would produce a harder baseline to beat. PfLDS is another example of a method that requires careful tuning, but nonetheless can work well. Of course, LFADS has many hyperparameters as wel... I also do not fully understand how LDA / PCA was applied in the rebuttal, but I think this is a potentially very useful and interesting comparison. Nonetheless, I will give the authors the benefit of the doubt that these comparisons were carefully done. Assuming that these new comparisons are included in the final version of the paper, and are carefully explained, I'm happy to see this paper accepted. My decision is borderline, however, so I'll leave the final decision up to the area chair. In the event that the paper is rejected, I encourage the authors to incorporate their revisions into the paper and resubmit elsewhere -- I think they greatly strengthen the work, and the paper will be a useful advance after those edits.


Review 3

Summary and Contributions: In this paper the authors describe how they modify a recently published method for inferring disentangled latent representations (GIN) so that it is suitable for neural population data. They apply the method to simulation experiments and two real datasets, showing that the obtained latent representations better correspond to the underlying tasks when compared to a regular VAE.

Strengths: The goal of obtaining interpretable latent representations for neural population data is of great interest and a very active domain of research at this moment. Specifically, in the VAE field, there has been a large number of publications on the topic of disentanglement, and identifying the correct methods for neural data is a pressing issue. The authors adopted a very recently proposed method and adjusted it for the application to neural data in a sound way.

Weaknesses: The authors don't quite succeed on giving convincing answers to pressing questions regarding their work. Mainly 1) Why their method should be chosen over popular alternatives for inferring population dynamics like LFADS (which is also VAE based). And 2) why the given method for obtaining disentangled latents is specifically suited for neural data. To address these points the authors would have to compare their method against stronger baselines (instead of just a vanillaVAE / tuning curve fitting). Furthermore, there is a lack of explanation and motivation for the GIN method. Without reading the relevant references it remains unclear why the conditioning on the labels or the volume-preserving generative model is needed. Furthemore the ground truth labels seem to play a central role for the method, but are not much discussed. The fact that latent dimensions are well separated for different conditions when they are used as input to the encoder seems trivial, so there should be more focus on other information that can be extracted from the latent space (like the analysis of the power spectrum in Fig. 4, there it is not clear however, if the same information can be found in the vanilla VAE latent space) or on the evaluation of pi-VAE without the latent prior.

Correctness: The methods seem to be correct.

Clarity: The paper is well written (with some small mistakes mainly in the introduction). The model lacks explanation of the applied methods. For example it does not become clear at all why the normalizing flow generative model is necessary. Furthermore the structure of the latent space z is not explicitly described (I assume it's a certain number of latents for every timestep).

Relation to Prior Work: As said before the proposed algorithm should be more clearly distinguished from and compared with other methods.

Reproducibility: Yes

Additional Feedback: ### Post author feedback I appreciate the work the authors put into performing those additional comparisons. I find them quite interesting, and with some additional details they definetely make this a stonger paper. I therefore raised my score to 6. To make this an even stronger paper the authors would have to come up with more convincing experiments where the variability we see in the latents not just corresponds to the prior input.

[Author Response · NeurIPS 2020]

We thank all the Reviewers for their constructive comments. By addressing all the major concerns, we strongly feel that the paper has significantly improved. Reviewer #1 is positive and had some clarification questions. Reviewer #2 and #4 seemed to raise conceptually the opposite objections: Reviewer #2 questioned the value of non-linear neural data analysis methods in general, while Reviewer #4 was more concerned that we didn't compare our pi-VAE to technically more complicated methods. Interestingly, these two sets of comments do converge to the same practical issue, which is the lack of comparisons to the alternative methods, which we now address (see Fig. 1). We will first address these major concerns by Reviewers #2 and #4, then turn to minor issues raised by Reviewer #1 and others.

Figure 1: (a-e) Reaching data. (a-d) Color-coded, averaged latent trajectories corresponding to each reaching direction was plotted for each method. The filled dot and cross represent starting and ending of the trial. (e) The first set of components for demixed PCA. (f-h) Hippocampus data.

Most significantly, we have analyzed the data using the methods suggested by the Reviewers, including linear (demixed PCA or dPCA, PCA, LDA+PCA) and nonlinear methods (supervised UMAP, PfLDS, LFADS). Whenever possible, the original authors' implementations of the methods were used. Fig. 1 summarizes some of these new results. **Reaching data:** Overall we find that, while the extracted latent structures from these methods exhibit interesting characteristics, none of them results in fully disentangled latents. Supervised UMAP recovers different directions as different clusters, but without clear representations of temporal dynamics. LFADS and PfLDS both lead to smooth trajectories. Although the trajectories for different directions are separated in the 4-dimensional space, directions and temporal dynamics are entangled so that it is difficult to interpret each individual latent dimension (Fig. 1b,c,d). dPCA with both time and directions as labels still entangles time and directions (stimulus components change with time) (Fig. 1e).

**Method considerations:** 1. LFADS can take task variables as external inputs to the model RNN. We thus try LFADS with or without reaching direction as external inputs (Fig. 1c,d). 2. dPCA only deals with discrete task variables each with the same number of trials and each trial with the same length, and is unable to recover additional latent fluctuations as our method. 3. UMAP can incorporate label information for supervised learning, and we use the reaching directions as labels (Fig. 1a). However, it doesn't recover temporal dynamics. **Take-away:** In latent space recovered by pi-VAE, dimensions separately encode temporal dynamics and reaching direction (as shown in our paper). In contrast, for latents learned by the alternatives, they are entangled and cannot be easily decoupled even after rotations (Fig. 1). Similar entangled results are found when applying these alternatives with 3-dimensional latents. Additionally, we find that pi-VAE could recover the geometry of the physical reaching targets, while other alternatives cannot. **Hippocampus data:** Applying the suggested methods (where supervised UMAP takes locations as labels), we find that the resulting latents are all less interpretable than pi-VAE (Fig. 1f,g,h), with no dimension directly representing the rat's location. Also the rhythmic-like fluctuations span across dimensions, rather than concentrate in one dimension (not shown). **Revision:** We will incorporate these plots and add detailed discussions on comparisons to the alternatives in revision, along with relevant references. These new results/comparisons substantially strengthen our conclusions, and they further highlight the identifiability and interpretability of latent representation recovered by pi-VAE. **Code availability**: Please be assured that the code will be documented and made available upon publication.

We would like to further note that these methods have different motivations and focuses, and how well they work in practice may depend on the questions being studied. Our method is motivated by leveraging the strength of regression-based methods and latent models to increase the identifiability and interpretability, a direction received little attention previously. To reduce the number of assumptions, we did not incorporate temporal smoothness priors, which are key for PfLDS, LFADS. Probably it would be best to consider these methods as complementary rather than competing methods. These points were briefly mentioned in the Discussion, and we will revise the text to make them more explicitly.

For questions raised by Reviewer #1: 1. Yes, we meant "inferred without label prior". 2. Yes, Poisson noise was used for VAE, and the only difference is that label prior was not used. 3. We completely agree that more emphasis should be put on difference between Fig. 3f and 3j, Fig. 4b and 4d. Thank you for this excellent suggestion. 4. The statement of "Poisson noise, continuous labels..." was made in reference to the specific implementation in the GIN paper. We agree that it was misleading. Finally, space limit prevents us from a discussion of the advantages of GIN here, please refer to their original paper. We will fix all these points, along with other minor concerns in the revised version.

[Meta-Review · NeurIPS 2020]

Reviewers, especially the most knowledgable, highlight a solid contribution. Paper is accepted.